# FLT3-ITD in Children with Early T-cell Precursor (ETP) Acute Lymphoblastic Leukemia: Incidence and Potential Target for Monitoring Minimal Residual Disease (MRD)

**DOI:** 10.3390/cancers14102475

**Published:** 2022-05-17

**Authors:** Luca Lo Nigro, Nellina Andriano, Barbara Buldini, Daniela Silvestri, Tiziana Villa, Franco Locatelli, Rosanna Parasole, Elena Barisone, Anna Maria Testi, Andrea Biondi, Maria Grazia Valsecchi, Carmelo Rizzari, Valentino Conter, Giuseppe Basso, Giovanni Cazzaniga

**Affiliations:** 1Cytogenetic, Cytofluorimetric and Molecular Biology Laboratory—Center of Pediatric Hematology-Oncology, Azienda Policlinico—San Marco, University of Catania, 95123 Catania, Italy; nellinaandriano@yahoo.it; 2Laboratory of Hematology-Oncology, Department of Woman and Child Health, University of Padova, 35128 Padova, Italy; barbara.buldini@unipd.it; 3Center of Biostatistics for Clinical Epidemiology, Department of Health Science, University of Milano-Bicocca, 20126 Milano, Italy; daniela.silvestri@unimib.it (D.S.); grazia.valsecchi@unimib.it (M.G.V.); 4Tettamanti Research Center, Department of Pediatrics, University of Milano-Bicocca, MBBM Foundation/ASST Monza, 20900 Monza, Italy; t.villa@gmail.com (T.V.); abiondi.unimib@gmail.com (A.B.); gianni.cazzaniga@asst-monza.it (G.C.); 5Department of Pediatric Hematology-Oncology, IRCCS “Bambino Gesù” Children’s Hospital, 00165 Rome, Italy; f.locatelli@opbg.net; 6Department of Pediatric Hemato-Oncology, A.O.R.N. Santobono-Pausilipon, 80122 Naples, Italy; r.parasole@santobonopausilipon.it; 7Pediatric Onco-Hematology, Regina Margherita Children’s Hospital, AOU Città della Salute e della Scienza, 10126 Turin, Italy; elena.barisone@unito.it; 8Department of Pediatrics, La Sapienza, University of Rome, 00100 Rome, Italy; testi@bce.uniroma1.it; 9Pediatric Hematology-Oncology Unit, Department of Pediatrics, University of Milano-Bicocca, MBBM Foundation/ASST Monza, 20900 Monza, Italy; c.rizzari@asst-monza.it (C.R.); valentino.conter@gmail.com (V.C.); 10Italian Institute for Genomic Medicine, 10100 Turin, Italy; giuseppe.basso@unipd.it; 11Department of Medical Genetics, School of Medicine and Surgery, University of Milano-Bicocca, 20126 Milano, Italy

**Keywords:** early T-cell precursor, ALL, children, FLT3-ITD, MRD

## Abstract

**Simple Summary:**

The prevalence of FLT3-ITD among children with ETP-ALL must be determined. MRD monitoring in ETPs is hampered by the lack of Immunoglobulin (IG) and T-cell receptor (TR) gene rearrangements. We determined the incidence of FLT3-ITD among children with ETP and performed MRD monitoring using FLT3-ITD sequences, successfully testing a new method of MRD detection. Moreover, we highlighted that the FLT3 pathway could represent a therapeutic target for precision therapy in patients with ETP.

**Abstract:**

Early T-cell precursor (ETP) is an aggressive form of acute lymphoblastic leukemia (ALL), associated with high risk of relapse. This leukemia subtype shows a higher prevalence of mutations, typically associated with acute myeloid leukemia (AML), including RAS and FLT3 mutations. FLT3-ITD was identified in 35% cases of adult ETP-ALL, but data in the pediatric counterpart are lacking. ETPs frequently lack immunoglobulin (IG) and T-cell receptor (TR) gene rearrangements, used for minimal residual disease (MRD) monitoring. Among 718 T-ALL enrolled in Italy into AIEOP-BFM-ALL2000, AIEOP-ALLR2006, and AIEOP-BFM-ALL2009 consecutive protocols, 86 patients (12%) were identified as ETP and 77 out of 86 children were studied for the presence of FLT3-ITD. A total of 10 out of 77 (13%) ETP cases were FLT3-ITD positive. IG/TR MRD monitoring was feasible only in four cases. FLT3-ITD MRD monitoring was performed using real-time PCR in all FLT3-ITD positive ETP cases. A comparison between IG/TR and FLT3-ITD resulted in comparable findings. Our study demonstrated that the FLT3-ITD prevalence in children was lower (13%) than that reported in adult ETP-ALL. FLT3-ITD can be used as a marker for sensitive molecular MRD monitoring in ETP-ALL when IG/TR markers are not available, potentially selecting those patients who should spare allogeneic hematopoietic stem cell transplantation (HSCT). Finally, the FLT3 pathway is a robust druggable target in this aggressive form of leukemia.

## 1. Introduction

Acute lymphoblastic leukemia (ALL) is an uncontrolled malignant proliferation of lymphoid cells in bone marrow (BM), peripheral blood (PB), and extramedullary sites. ALL is the most frequent neoplasia in childhood, and it still represents the first cause of death in children with cancer [1]. This malignant neoplasm is usually presented as B-cell precursor ALL (BCP-ALL), which accounts for 85% of childhood leukemias [1]. T-cell acute lymphoblastic leukemia (T-ALL) is less frequent than BCP-ALL, and it is characterized by heterogeneous and variable genetic abnormalities in T-lymphoid cells. T-ALL tends to occur more often in adult than in pediatric patients, showing an incidence of 25% and 15%, respectively [2]. T-ALL has an unusual morphology and genetic and clinical features, characterized by male sex, leukocytosis, rapid infiltration, and a median age of 9 years. Almost 50% of patients present with a high white blood cell count (or hyperleukocytosis) [3]. In the majority of T-ALL cases (60%), a mediastinal mass is frequently observed and about 10% of cases have a predisposition to central nervous system (CNS) involvement [3]. Although in the past outcomes for children with T-ALL were inferior to B-ALL, current higher intensive chemotherapy protocols induce nearly equivalent prognoses for childhood T-ALL and B-ALL, respectively [4,5].

Morphology, immunophenotype, and identification of genetic abnormalities represent the standard procedures for making a diagnosis of ALL [5]. Immunophenotyping is essential for an ALL classification. Lymphoid T-cell precursors are assigned to different categories, based on the European Group of Immunological Characterization of Leukemia (EGIL): the immature pro-T (T-I), pre-T (T-II), cortical T (T-III), mature-T (T-IV), and T-γ/δ [6]. The pro-T and pre-T gene expression patterns are associated with better outcomes than cortical or medullary types [3]. In 2016, the World Health Organization (WHO) classification of ALL included a new subtype, early T-cell precursor ALL (ETP-ALL), which represents 15% of T-ALL cases [4]. The gene expression and immunology of a new subtype were first described in 2009 [7]. ETP-ALL is characterized by immature precursors related to hematopoietic stem cells and myeloid progenitors [7,8,9]. Early diagnostic criteria for the definition of ETP were those established by St. Jude [7] and further applied by other cooperative groups [10,11,12]: lack of CD1a and CD8 expression, the absence or weak expression of CD5, and the positivity (>20%) of at least one between myeloid (CD13, CD33, Cd11b, CD65) and/or stem cell (CD34, CD117,) markers [7,10,12]. 

In the early original study, ETP-ALL was associated with a poor prognosis [7]. However, more recent approaches demonstrated that ETP might have an outcome similar to non-ETP in a pediatric setting [12]. T-ALL is characterized by heterogenous genomic alterations: transcriptional activation of several protooncogenes, deletions of tumor suppressor genes, epigenetic deregulation, ribosomal dysfunction, altered RNA stability, cell-cycle dysregulation, and disordered signaling in the crucial pathways (NOTCH1/FBXW7, PI3K/Akt/mTOR, RAS/MAPK, and IL7R–JAK–STAT) [13]. In ETP-ALL, the presence of different gene-mutations involving hematopoietic development (*IKZF1, ETV6, RUNX1, GATA3,* and *EP300*), MAPK and cytokine receptor signaling (*FLT3*, *NRAS, KRAS, IL7R, JAK1, JAK3, PTPN11, NF1*, and *SH2B3*), and chromatin-modifying genes (*EED, EZH2, SUZ12*, and *SETD2*) have been demonstrated [2]. Inactivation by the mutation or deletion of *ETV6, RUNX1*, and *GATA3* has also been described in patients with Acute Myeloid Leukemia (AML) and correlates with poor outcomes in ETP-ALL [4,8,14].

Consistently, in this leukemia subgroup a higher prevalence of mutations was detected that are typically associated with AML, including *FLT*3 mutations [8,15]. FLT3-ITD was identified in up to 35% of adult ETP-ALL [15], but data on its prevalence in pediatric ETP-ALL are lacking. Moreover, FLT3-ITD represents a therapeutic target for several new compounds in AML, showing improvement in survival rate when associated with conventional chemotherapy [16]. The measurement of minimal residual disease (MRD) has deeply modified the risk stratification in the treatment of pediatric ALL, applying different strategies [5,17]. However, consistently with its stem cell signature, ETPs frequently lack IG and TR gene rearrangements, the most used and sensitive targets for MRD monitoring. Therefore, alternative markers are required to extend the application of molecular MRD to most ETP-ALL patients. In particular, specific efforts are mandatory and aim to identify the cases who need to be redirected or not to an allogeneic stem cell transplantation (Allo-HSCT) and/or to a targeted therapy with FLT3 or other specific pathway’s inhibitors, since this form of leukemia showed a high plasticity and chemoresistance at relapse [18]. We explored the prevalence of *FLT3-ITD* mutation in a large series of pediatric ETP-ALL enrolled in three consecutive protocols of the Italian Association of Pediatric Hematology and Oncology (AIEOP) and we evaluated the potential use of *FLT3-ITD* as an alternative DNA marker for MRD monitoring.

## 2. Materials and Methods

In our study, we evaluated 718 T-ALL cases enrolled in Italy into the AIEOP-BFM-ALL2000, AIEOP-ALLR2006, and AIEOP-BFM-ALL2009 consecutive protocols between 2000 and 2016. ETP characterization was performed by application of well-defined protocols by flow cytometry (FCM): criteria for definition of ETP were those established by the AIEOP-BFM Flow Network [10,12].

All cases enrolled in these protocols were screened for the presence of *BCR-ABL1* fusion transcript and *KMT2A* rearrangements. ETP-ALL cases of our cohort resulted negative for such molecular screening.

We performed Ficoll gradient centrifugation and cryopreservation of mononuclear cells from patients’ BM samples. DNA was extracted by using 5 PRIME kit (Roche Diagnostics, Basel, Switzerland) from cryopreserved BM or PB diagnostic cells, following the manufactures instructions. The presence of FLT3-ITD in all patients at diagnosis was detected by PCR on genomic DNA, using specific forward (11 F) and reverse (12 R) primers; the mean length of the ITD was 44 nucleotides (nts) (range 24–71), with a mean of 7 randomly inserted nts (range 1–26). Aberrant PCR products were cloned into the pMOSBlue vector (pMOSBlue Blunt Ended Cloning Kit, Amersham Pharmacia Biotec, Buckinghamshire, UK). Plasmidic DNA from recombinant colonies was sequenced on an ABI PRISM 377 Automated Sequencer (Applied Biosystems, Foster City, CA, USA) [19].

In order to detect the MRD using FLT3-ITD sequences, a patient-specific forward primer, a probe, and a reverse primer were designed using Primer Express software, version 3.0 (Applied Biosystems, Foster City, CA, USA). We designed the patient-specific forward primer in order to preferentially anneal to the joining region, including the wild-type sequence and the beginning of the FLT3-ITD region. We also designed two different common reverse primers and one fluorescent probe in order to reduce the risk of any duplication (Appendix A). For RQ-PCR analysis, the TaqMan PCR core reagent kit was used (Applied Biosystems, Foster City, CA, USA). Real-time analysis was performed on the ABI PRISM 7700 or 7900 Sequence Detection System containing a 96-well thermal cycler (Applied Biosystems, Foster City, CA, USA). The RQ-PCR was optimized for each patient by changing the annealing temperature in order to increase sensitivity. All samples were tested in triplicate. To test sensitivity and specificity of the assay, 10-fold serial dilutions were prepared, starting from 500 ng of diagnostic BM-MNC DNA mixed with DNA extracted from a pool of five different healthy donors. To correct for the quantity and quality of DNA in follow-up samples, the albumin gene DNA was amplified (TaqMans Control Genomic DNA, Applied Biosystems). Standard curves for FLT3-ITD and albumin were established by amplifying a 10-fold serial dilution of target DNA in DNA from healthy donors PB-MNC or water, respectively. Subsequently, the albumin-normalized diagnostic value was set as 1.0 and FLT3-ITD-normalized specific quantities in follow-up samples were related to that at diagnosis [20].

Details on probe design and MRD analyses using FLT3-ITD sequences are shown in Appendix A. MRD analyses using IG/TR on the same sample and flow cytometry-MRD were performed in parallel by standard procedures [21]. EuroMRD guidelines were applied for performance and interpretation of RQ-PCR [22].

We conducted this study according to the guidelines of the Declaration of Helsinki. The study was approved by local Ethical Committee of each AIEOP center, applying the cited protocols.

## 3. Results

We identified 86 cases with ETP. Among these patients, 77 were screened for the presence of FLT3-ITD mutation. Characteristics of all patients with ETP are listed in Appendix A. We found 10 cases with FLT3-ITD (13%). Characteristics of these children are listed in Table 1.

We examined 10 ETP patients FLT3-ITD positive, 9 male and 1 female; the average age at diagnosis was 9 years, ranging between 1–6 and 17 years. The majority of cases presented with a white blood cell count at diagnosis ranged between 20.000 and 100.000/μL, with a percentage of blasts ranging between 60–90%. After the pre-phase with prednisone, we found two cases with a good prednisone response (PGR), whereas eight children presented with a poor response (PPR), as shown in Table 1. In our previous clinical study, ETP-ALL was diagnosed in 49 patients, characterized by the absence of molecular markers for PCR detection of MRD in 25 (56%) out of 45 patients (2). FLT3-ITD MRD monitoring was performed in all 10 cases with FLT3-ITD positive ETP-ALL (Appendix A). Median length of the ITD region was 55.5 nucleotides (nts) (range 26–254), with a median of 4 randomly inserted nts (range 1–21). Standard curves performed by 10-fold dilutions in DNA from peripheral blood (PB) of healthy donor (HD) showed a quantitative range (QR) of 5 × 10^−4^ in all cases and 1 × 10^−4^ in 3 out of 10 and 1 × 10^−3^ in one case, respectively. Sensitivity of the assay was at least 1 × 10^−4^ in 2 out of 10 cases and 1 × 10^−5^ in the remaining cases (Appendix A). FLT3-ITD MRD monitoring was also performed in four cases with FLT3-ITD positive non-ETP ALL. Median length of the ITD was 33 nts (range 24–63), with a median of 4 randomly inserted nts (range 3–17). Standard curves performed by 10-fold dilutions in DNA from PB-HD showed a quantitative range of 5 × 10^−4^ in two cases and 1 × 10^−4^ in the other two cases. Sensitivity of the assay was 1 × 10^−5^ in two and 1 × 10^−4^ in the other two cases, respectively (Appendix A). A comparison between IG/TR and FLT3-ITD was feasible in three cases (Appendix A). We monitored all three cases by two TR markers. We show here the comparable findings in ETP-Case 4 (Figure 1A) and in ETP-Case 9 (Figure 1B). At day15 and day33 during induction therapy, when MRD was very high (ranged between 1 × 10^−1^ to 1 × 10^−3^), the TR and FLT3-ITD were almost fully comparable. At day78 (end of induction—phase IB) in ETP-Case 4 (Figure 1A) and ETP-Case 9 (Figure 1B) the MRD analysis resulted fully negative for both markers. In ETP-Case 3 (Figure 1C) the kinetic of MRD reduction was detected by all three markers, although the Quantitative Range was lower than the minimum allowed by Euro-MRD guidelines [21].

## 4. Discussion

For the first time to date, our study reported the FLT3-ITD prevalence in a consecutive series of children with ETP-ALL, which resulted to be 13% (10/77), a value lower than that reported in adult ETP-ALL [15]. FLT3 mutations in children with T-ALL have been reported already [23]. Furthermore, this genetic lesion was found to be highly recurrent in ETP by applying the whole genome sequencing (WGS) on 12 pediatric cases, in association with other pathways’ aberrations, indicating that ETP is a stem cell leukemia [8]. Moreover, several pieces of evidence demonstrated that in ETP cells, specific targetable pathways as FLT3, PI3K-AKT and Bcl-2 are considered suitable therapeutic targets [18,24,25].

Modern protocols for the treatment of ALL are fully based on detection of MRD along the induction phase [17]. It has been clearly demonstrated that the reduction curves of MRD are different in T-ALL and B-ALL [26]. In fact, although MRD was detectable in the majority of T-ALL patients at the end of induction, they still had a favorable outcome when they showed a low level or undetectable MRD at the end of consolidation, or phase I-B [26,27]. Consistently with its stem cell signature, ETPs frequently lack IG/TR gene rearrangements, the most used and sensitive targets for MRD monitoring. This inconvenience occurred in almost 60% of cases with ETP [12]. For this reason, we investigated the feasibility of MRD detection using FLT3-ITD. We found a potential marker in most cases, showing a strict concordance with TR based MRD methodology (Figure 1), even in non-ETP-ALL with FLT3-ITD (Appendix A). However, in case of discrepant results in comparison with IG/TR, we point out that detection of MRD using FLT3-ITD could result negative. This discrepancy could be explained by the presence of FLT3-ITD mutation in a minority of blast cells as secondary mutation, which might be more sensitive to the treatment. Nevertheless, although in a single case, a next generation sequencing (NGS) strategy for detecting MRD in ETP-ALL has been recently applied and was potentially effective in association with WGS analysis for identification of multiple target genes. It is reliable to then estimate the subclonality and to identify an appropriate target for MRD monitoring [28].

Moreover, using integrated genomic analysis, it has been demonstrated that FLT3 and JAK-STAT activations are frequently found in T-cell immature/ETP ALL, suggesting a targeted therapy [14]. In our cohort of patients, FLT3-ITD was detected in 13% of cases, but the role of this aberration in ETP-ALL is still unclear. Nevertheless, FLT3-ITD, as an MRD marker, should be followed in children with ETP-ALL during the first three months of therapy, reducing the risk of early disappearance for a clonal shift. In AML, FLT3-ITD is a driver mutation, conferring a poor prognosis and mostly (75%) re-appearing at relapse [29]. This type of aberration could be detected as a higher allelic ratio in respect to the wild type allele [29]. Moreover, a previous study reported that exposure to FLT3 inhibitors induced high rates of apoptosis in FLT3 high-expressed T-lineage leukemic cells [30].

It has been also demonstrated that components of polycomb repressive complex 2 (PRC2) as epigenetic regulators (*SUZ12*, *EED*, and *EZH*2) are frequently mutated or deleted in ETP-ALL cases [8]. FLT3 resulted in significantly upregulated PRC2 mutated leukemic cells after an RNA sequence profiling. This finding was surprisingly confirmed by the prolonged survival of *EZH2* mutated T-ALL cells in mice treated with sorafenib and quizartinib [24]. This combination could be applied in immature/ETP patients.

Remarkably, other researchers have discovered, through BH3 profiling, that ETP-ALL is Bcl-2 dependent and is very sensitive to in vitro, testing an ETP cell line (LOUCY) and in vivo treatment with ABT-199, a drug tolerated well in clinical trials [25]. Based on this experience, a recent clinical application demonstrated that Venetoclax (ABT-99) combined with chemotherapy was effective, especially in patients with ETP, resulting in a MRD negativity and a long-term survival [31]. For this reason, it is conceivable to introduce a specific drug during first-line treatment for those cases who presented with a high rate of MRD at the end of induction phase.

Therefore, based on our findings and recent novel approaches [9,28], it is mandatory in this rare subgroup of patients to perform extensive NGS-based genomic analyses to monitor MRD and to identify multiple target genes for a more innovative and effective treatment.

We demonstrated here that FLT3-ITD could be used as a marker for sensitive molecular MRD monitoring in ETP-ALL when IG/TR markers are not available, applying a well-established low-cost methodology. Our results of MRD monitoring at day 78 (or Timepoint 2) confirmed that ETP patients respond well to consolidation therapy or Phase I-B [12,27], although the introduction of FLT3 or JAK-STAT inhibitors has been recently applied in that specific phase of treatment (NCT03117751—ClinicalTrials.gov).

Moreover, since ETP has a peculiar immunophenotype, MRD analyses should be performed applying both FCM and PCR methodologies in a synergistic way. This could reduce the false negative because FLT3-ITD is a secondary mutation, which could not be present in all the leukemic cells sometimes.

## 5. Conclusions

In conclusion, the limitations of our study (retrospective analysis, a small number of cases, and a lack of comparison to FCM methodology for MRD detection) do not allow us to draw any statement on the prognostic impact of FLT3-ITD in ETP-ALL, although 8 out of 10 patients are alive in complete continuous remission (CCR). However, we are not so far able to consider FLT3 pathway as a potential therapeutic target in cases with ETP in association with other tyrosine kinase or JAK-STAT inhibitors [14,24,30] or Bcl-2 inhibitors [31], as well as in AML [29,32]. Moreover, considering that the ETP cases are allocated into the high-risk arm, the MRD detection using FLT3-ITD in association with FCM methodology as well as in children with AML [33] should allow us to characterize those cases for whom the hematopoietic stem cell transplantation could be avoided or chosen wisely, thus introducing novel drugs with promising efficacy along the induction phase or even the entire duration of the treatment plan. Therefore, a prospective study is necessary.

## Figures and Tables

**Figure 1 cancers-14-02475-f001:**
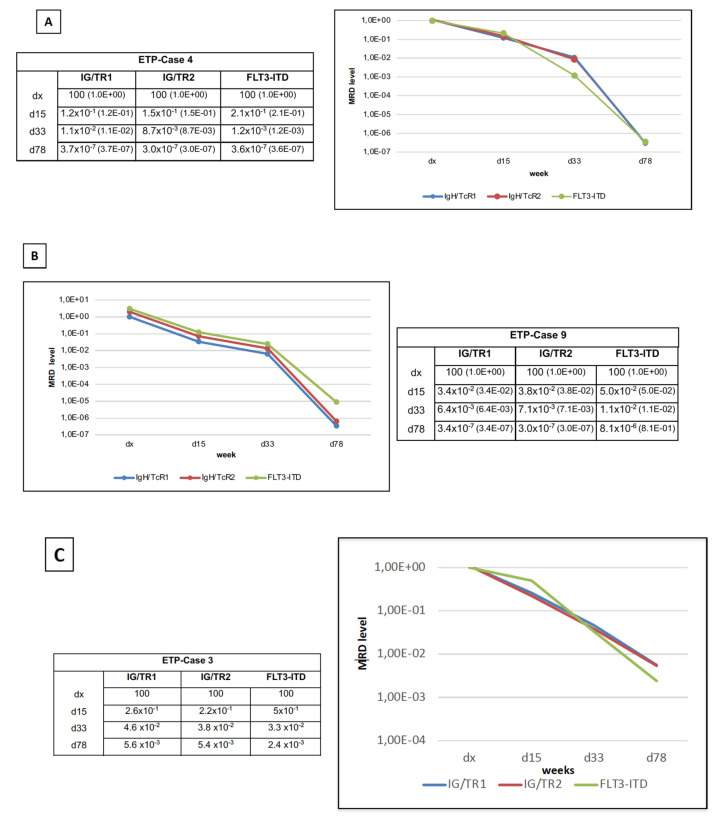
Diagrams of MRD levels in three children with ETP-ALL, comparing the IG/TR with FLT3-ITD as markers of detection. Analyses were performed during the induction phase, as well as at day15, day33, and day78, respectively. (**A**,**B**) MRD levels of patients in ETP-Case 4 and ETP-Case 9 almost completely overlapped in all three consecutive time points. In ETP-Case 3 (**C**) the kinetic of MRD reduction was detected by all three markers, although the Quantitative Range did not reach the 1 × 10^−4^.

**Table 1 cancers-14-02475-t001:** Characteristics of children with FLT3-ITD positive ETP, enrolled in AIEOP protocols.

**Total Patients**		*n°* 10	100%	**Total Patients**		*n°* 10	100%
**Age**	1–5 yrs	2	20	**N° IG/TR markers**	0	6 *	60
	6–9 yrs	3	30		1	1	10
	10–17 yrs	5	50		2	3	30
**Gender**	Male	9	90	**PCR MRD Risk**	Standard	0	0
					Medium	2	20
	Female	1	10		High	2	20
					Not Known	6 *	60
**WBC**	<20,000	1	10	**FCM MRD Risk**	Standard	0	0
	20–100,000	7	70		Medium	2	20
	≥100,000	2	20		High	4	40
					Not Known	4 ^	40
**PDN response**	Good	2	20	**Final Clinical Risk**	Standard	0	0
	Poor	8	80		Medium	0	0
					High	10	10
**CR-Ia**	CR	9	90	**Outcome**	Death in CCR	0	0
	Resistant	1	10		Death after HSCT	2	20
					Alive in CCR	8	80

Yrs years; WBC white blood cell; PDN prednisone; CR-Ia complete remission after induction phase Ia; IG/TR Immunoglobulin/T-cell receptor; PCR polymerase chain reaction; FCM flow cytometry; CCR continuous complete remission; HSCT hematopoietic stem cell transplantation; * Cases with ETP-ALL with no markers and no MRD analysis. ^ Cases with ETP-ALL enrolled in AIEOP LLA-2000 and R-2006 were not prospectively studied for the detection of MRD by FCM at day+15.

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
