# Peer review of "FLT3-ITD in Children with Early T-cell Precursor (ETP) Acute Lymphoblastic Leukemia: Incidence and Potential Target for Monitoring Minimal Residual Disease (MRD)"

_cancers, 2022, doi:10.3390/cancers14102475_

Round 1

Reviewer 1 Report

The authors have analyzed the genetic marker FLT3-ITD which can be used as a marker for sensitive molecular MRD monitoring in ETP-ALL, when IG/TR markers are not available, potentially selecting those patients who should spare allogeneic hematopoietic stem cell transplantation (HSCT). In my opinion, it is a very interesting and important study because, in recent years, a subgroup of early T-cell precursor (ETP)-ALL have been identified among patients, which is characterized by worse prognosis and chemoresistance in comparison to non-ETP-ALL. In terms of immunophenotype, chromosomal aberrations, and genetic mutations, ETP-ALL is more similar to a myeloid malignancy, therefore there may be a need to separate this group from all T-ALL cases in order to apply another targeted treatment. Therefore, each study for his group of patients is so important from the clinician's point of view. 

Minor revision.

  1. No genetic prognostic factor has been found for T-ALL so far in the classification of risk group. Therefore, T-ALL is the subject of intensive research among scientists. But I think that basic genetic tests should be performed for all T ALL patients. There is a lack table which would contain basic genetic tests: BCR/ABL1, KMT2A, or karyotype or more specific LMO2/LYL1, HOXA, BCL2, or NUP98 or karyotype. 

Author Response

Dear Reviewer, #1,

on behalf of the authors, I am grateful for your comments and appreciation of the manuscript. 

As reply to your comment, we performed a retrospective analysis using the left-over materials, thus we concentrated our analysis specifically on the detection of FLT3-ITD. Moreover, children with T-ALL enrolled into AIEOP-ALL protocols have been studied for the presence of BCR-ABL1 fusion transcript and KMT2A rearrangements at diagnosis. Please check row 123.

Karyotype analyses have not been centralized and for this reason in the majority of the cases we missed this information. The other genes that you suggested, were usually studied at relapse, in a wide genomic fashion analysis, for research purpose.  

Reviewer 2 Report

First the paper is interesting but the quality of the Table could be improved (one column for the N and one for the %).

The population of FLT3mutated patients could be compared to the FLT3wt patients to know whether this group of patients share common characteristics distinct from other patient (higher blood cell counts for instance).

Authors see a reduction in the FLT3 mutation load in two patients (Figure 1), but they performed the test for a third one. It appears that they could not detect the mutation as displayed in the supplementary material. Can the authors comment about this point?

Finally, about 25% of AML patients loose FLT3 mutation at relapse. As a consequence, FLT3-ITD is not a suitable marker in AML. Could it be the case in ALL as well. It raises concerns about the suitability of FLT3 mutation for detecting relapse in ALL setting. Could the authors discuss this point?

Author Response

Dear Reviewer, #2,

on behalf of the authors, I am grateful for your comments and appreciation of the manuscript. 

As reply to your comments:

1-) We implemented the presentation of Table 1, following your suggestions.

2-) We inserted a new Supplementary Table 1, comparing all the data between cases with ETP and FLT3-ITD vs cases with No FLT3-ITD, respectively.

3-) We implemented Figure 1, including the ETP-Case 3 showing a lower quantitative range of MRD detection.

4-) We completely agree with this statement: FLT3-ITD is not so stable in AML, thus it should be as well as in ETP. For this reason, we strongly suggest to associate flow cytometry to FLT3-ITD PCR technology in order to detect MRD in children with ETP as well as we already performed in AML. Consistently, we inserted a reference to support this concept [33] (Buldini B et al.).

Reviewer 3 Report

In this manuscript, Lo Nigro and colleagues point out the possible potential of FLT3-ITD to be used for MRD monitoring in ETP ALL. Although the few number of patients, the results are clearly presented and interesting. Methodology seems also correct and well described. The idea of monitoring ETP lacking covers an unmet need. However, I consider the length of the manuscript could better be a Letter to the Editor because new data is limited. In addition, I have few comments that could be included/discussed in the Discussion Section and are shown below.

- Thank you, it is nice to know FLT3-ITD frequency in children.

- Line 120, the sentence“Method of ETP characterization by FCM” could be more formally written.

- ITD in ETP ALL cases seems to be significantly larger than those observed in non-ETP ALL. Do you have any explanation for this? Do you think that larger duplications have different functional implications (i.e. higher pathway activation?)

- Please, a comment on discussion could be done for:

1) FCM could be used for monitoring MRD for FLT3-ITD ALL but also for all cases of ETP ALL (more widely applicable than FLT3-ITD monitoring), specially ETP having this special phenotype.

2) Stability of FLT3-ITD for MRD monitoring can be somewhat questionable, as seen in AML. FTL3-ITD is a secondary genetic alteration (not necessarily present in all tumour cells) and monitoring 2ary alterations could be risky because this method may not track leukemia stem cells negative for FLT3-ITD.

Author Response

Dear Reviewer, #3,

on behalf of the authors, I am grateful for your comments and appreciation of the manuscript. 

We are glad that you highlight the basic novelty of our study: incidence of FLT3-ITD among children with ETP.

As reply to your comments:

1-) Please check the new version of ETP characterization with FCM on rows 120-123.

2-) The difference of ITD length between ETP and non-ETP ALL could be related to the immature phenotype of the former group.

3-) Discussion comments 1 and 2: we performed a retrospective analysis which did not include an MRD detection by FCM. For this reason, we erased the sentence in the abstract (row 45 “as well as flow-cytometry”) and we inserted a specific recommendation to use FCM in association with PCR technology for the detection of MRD as well as we performed among children with AML (see the new reference [33] Buldini B et al).

Reviewer 4 Report

The authors report the results of the use of FLT-ITD as an MRD target in 77 ETP-ALL patients enrolled in three consecutive AIEOP treatment protocols. Ten (13%) cases of the ETP-ALL were positive for the FLT-ITD. Six out of ten FLT-ITD-positive patients did not have a suitable IgG/TCR-specific MRD target, ie. these patients relied on the flow cytometry for therapy response assessment. The main finding is that in these cases, FLT-ITD could serve as a supplementary marker to help decide which patients need therapy intensification and perhaps even allo-HSCT. This clearly has a significant bearing on the patients therapy choices.

Overall, the study is relevant and well-performed. The downside is the low number of FLT-ITD cases, but this is understanable given the rarity of ETP-ALL in children and the fact that only a minor proportion of cases are positive for FLT-ITD. I have some minor suggestions to authors for consideration.

1) The authors are referring to the flow cytometry results (row 45) in the abstract but they are missing in the results section. I would suggest adding them to Figure 1 (see also comment #6).

2) I suggest the authors to show also the data for the third (and fourth?) case in whom both FLT-ITD and IG/TCR markers were available in Figure 1.

3) Table 1 is lacking the legend. Moreover, the layout of Table 1 somewhat peculiar, so I would like the authors to consider more traditional layout, and perhaps including mean/median and spread of values (range/SD/IQR) in the Table.

4) Sentence 182-183: It should perhaps be written in the opposite order, for example: "Sensitivity of the assay was at least 1.0E-04 in 2 out of 10 cases, and 1.0E-05 in the rest".

5) The authors are using mean and range in the text to describe the length of ITD/inserted nts. I would prefer using median and range as the distribution seems not to be normal.

6) Figure 1. Remove the word "Legend" in figure legend. Consider improving the layout: e.g. putting the PCR measurement values below the graph at right time points (similar to survival curves where patients at risk are indicated below curves).

7) Discussion rows 220-222. I think that this is too bold statement as you only had three cases with concomitant FLT-ITD and IgG/TCR markers. Unless you are referring here to flow cytometry data that is not shown in the manuscript

8) Rows 226-230. Difficult sentence, please cut in pieces or improve otherwise.

Author Response

Dear Reviewer, #4,

on behalf of the authors, I am grateful for your comments and appreciation of the manuscript. 

As reply to your comments:

1-) we performed a retrospective analysis which did not include an MRD detection by FCM. For this reason, we erased the sentence in the abstract (row 45 “as well as flow-cytometry”) and we inserted a specific recommendation to use FCM in association with PCR technology for the detection of MRD as well as we performed among children with AML (see the new reference [33] Buldini B et al).

2-) We implemented Figure 1, including a third case, although with lower quantitative range of sensitivity.

3-) We implemented Table 1 and we inserted a new Supplementary Table 1 with all the information regarding cases with ETP.

4-) We modified the sentence: see row 189.

5-) We modified the Supplementary tables (see Table 2 and 3) and the text (see rows 185-186 and 191-192).

6-) We removed the “Legend” in Figure 1.

7-) We suggested that FLT3-ITD could be considered as Potential Marker of MRD in association with other more reliable methodologies as NGS and Flow-cytometry. We are referring to T-lineage ALL in general.

8-) We dissected the entire sentence, in order to clearly expose our comments.

This manuscript is a resubmission of an earlier submission. The following is a list of the peer review reports and author responses from that submission.